# Influence of Specific Interventions on Bracing Compliance in Adolescents with Idiopathic Scoliosis—A Systematic Review of Papers Including Sensors’ Monitoring

**DOI:** 10.3390/s23177660

**Published:** 2023-09-04

**Authors:** Claudio Cordani, Lia Malisano, Francesca Febbo, Giorgia Giranio, Matteo Johann Del Furia, Sabrina Donzelli, Stefano Negrini

**Affiliations:** 1Department of Biomedical, Surgical and Dental Sciences, University “La Statale”, 20122 Milan, Italy; claudio.cordani@unimi.it (C.C.); stefano.negrini@unimi.it (S.N.); 2IRCCS Istituto Ortopedico Galeazzi, 20157 Milan, Italy; matteo.delfuria@grupposandonato.it; 3Postgraduate School of Physical Therapy and Rehabilitative Medicine, University “La Statale”, 20122 Milan, Italy; liamalisano@gmail.com (L.M.); giorgia.giranio@gmail.com (G.G.); 4ISICO (Italian Scientific Spine Institute), 20141 Milan, Italy; sabrina.donzelli@isico.it

**Keywords:** scoliosis, spinal diseases, braces, orthotic devices, electrical equipment and supplies, patient compliance

## Abstract

Adolescent idiopathic scoliosis (AIS) is a common disease that, in many cases, can be conservatively treated through bracing. High adherence to brace prescription is fundamental to gaining the maximum benefit from this treatment approach. Wearable sensors are available that objectively monitor the brace-wearing time, but their use, combined with other interventions, is poorly investigated. The aims of the current review are as follows: (i) to summarize the real compliance with bracing reported by studies using sensors; (ii) to find out the real brace wearing rate through objective electronic monitoring; (iii) to verify if interventions made to increase adherence to bracing can be effective according to the published literature. We conducted a systematic review of the literature published on Medline, EMBASE, CINAHL, Scopus, CENTRAL, and Web of Science. We identified 466 articles and included examples articles, which had a low to good methodological quality. We found that compliance a greatly varied between 21.8 and 93.9% (weighted average: 58.8%), real brace wearing time varied between 5.7 and 21 h per day (weighted average 13.3), and specific interventions seemed to improve both outcomes, with compliance increasing from 58.5 to 66% and brace wearing increasing from 11.9 to 15.1 h per day. Two comparative studies showed positive effects of stand-alone counseling and information on the sensors’ presence when added to counseling. Sensors proved to be useful tools for objectively and continuously monitoring adherence to therapy in everyday clinical practice. Specific interventions, like the use of sensors, counseling, education, and exercises, could increase compliance. However, further studies using high-quality designs should be conducted in this field.

## 1. Introduction

Adolescent idiopathic scoliosis (AIS) is a progressive condition that affects 2–3% of adolescents aged 10–16, with a greater prevalence in females. It consists of a three-dimensional deformity of the spine and trunk [1]. The progression rate is quite variable, with rapid growing phases representing the most delicate moment for curve progression [2]. While mild curves may only result in aesthetic issues, more severe curves may cause adulthood back pain and more severe issues, such as spinal unbalance and camptocormia, leading to increased disability and an impact on quality of life [3,4].

Bracing is the most widely used conservative approach to AIS. Its purpose is to reduce the curve and/or prevent its progression, especially during the growth spurt phase, which represents the highest risk phase. Braces designs may vary, as may the materials, ranging from soft to rigid braces [5]. The effectiveness of brace treatment in affecting the progression of spine deformity is still debated, especially when it comes to more severe curves [6], with surgery being the most common alternative approach. Nevertheless, it is widely recognized that even though brace treatment may not improve curves and reduce spine deformity, it can positively affect the natural course of scoliosis progression [1,6].

Brace-wearing hours regimes vary depending on factors such as curve severity, age, and bone maturity. The BrAIST randomized controlled trial (RCT) showed that the more frequently the brace is worn, the better the results [7]. Unfortunately, patients included in the BrAIST study seldom used the brace for more than 18 h per day. The comparison with the ISICO database confirmed this result, showing this also in patients wearing the brace for more than 18 h per day [8]. Consequently, therapeutic outcomes are tremendously affected by adherence to the prescribed regimen of wearing hours [1,9]. Many factors were found to be responsible for negatively influencing compliance; some of these factors are patient related, such as age and psychological features [10]. Other aspects that hinder compliance are associated with the side effects of bracing, such as discomfort, restricted movement, and a negative cosmetic appearance [11].

Available review articles investigating different adherence factors in AIS [10,11] identified two major technologies used in electronic monitoring: temperature-based and force-based systems. These studies indicate the use of electronic devices to be the most promising approach to measuring compliance with bracing, since it appeared that common methods, like diaries, questionnaires, and confrontations with parents, can lead to the overestimation of wearing hours [10]. However, the authors performed a careful interpretation of these findings in light of the limited high-quality evidence, which precludes reliable conclusions.

The objectives of this study are (1) to summarize the real compliance with bracing reported by all studies that specifically used sensors, (2) find out about real brace wearing through objective electronic monitoring, and (3) verify if interventions performed to increase adherence to bracing can be effective according to the published literature.

## 2. Materials and Methods

### 2.1. Design

We performed a systematic review of the compliance to the bracing treatment in AIS, which was objectively measured through electronic sensors. The review was conducted following the Preferred Reporting Items for Systematic Reviews and Meta Analyses (PRISMA) guidelines [12]. The protocol was registered on PROSPERO (CRD42023427906).

### 2.2. Study Selection

#### 2.2.1. Search Strategy

An information specialist (MJDF) searched for papers from the study’s inception until 22 April 2023 using the following databases: Medline, EMBASE, Cumulative Index to Nursing and Allied Health Literature (CINAHL), Scopus, Cochrane Central Register of Controlled Trials (CENTRAL), and Web of Science.

We also searched the reference lists of any relevant systematic reviews published within the search period.

#### 2.2.2. Type of Studies

We included observational (i.e., cross-sectional, case–control, retrospective, and prospective cohorts) and interventional studies that aimed to monitor the usage of braces in the treatment of adolescents affected by idiopathic scoliosis through sensors. Case series were also included. We excluded case reports, poster abstracts, expert opinions, letters to the editor, non-English-language or unavailable full texts, and all studies that did not meet the inclusion criteria.

#### 2.2.3. Population

The study population included patients under 18 years of age who were affected by AIS with any Cobb angle and Risser stage and treated with a brace intervention of any kind.

#### 2.2.4. Interventions

We included studies performed using sensors of any kind that aimed to monitor the usage of braces in the treatment of adolescents affected by idiopathic scoliosis. The sensors could measure temperature, movement, or forces, and they could evaluate the quality of brace wearing or monitor the activity of patients during brace wearing. All studies performed to monitor brace treatment adherence in patients affected by other diseases affecting the spine were not considered to be eligible and were, consequently, excluded.

We classified the studies according to the presence of any intervention made to increase compliance, such as education, cognitive–behavioral approaches, information regarding the presence of the sensors, etc.

#### 2.2.5. Comparators

We considered four kinds of comparator: control, usual care, sham, and no intervention. In addition, studies with a single group were considered.

#### 2.2.6. Outcomes

We considered the following factors: (1) the time spent by the participants wearing the brace, as registered via the sensors as the primary outcome; (2) compliance with the prescribed wearing regimen of bracing as the ratio of the number of wearing hours registered via sensors to the number of hours prescribed by the clinician; (3) the reported adherence as the ratio of the number of hours of wearing reported by the patients to the number of hours prescribed.

We used reported data to calculate this information if it was not directly reported within the paper.

### 2.3. Study Screening

Four reviewers (CC, FF, LM, GG) independently performed the screening of the title and abstract retrieved using the search strategy and assessed the full-text articles for potential inclusion. Disagreements were resolved through discussion with two other authors (SD, SN) until a consensus was reached. Rayyan software (https://www.rayyan.ai, (accessed on 1 August 2023)) was used to manage these phases.

### 2.4. Data Extraction

Two reviewers (LM, GG) independently extracted general characteristics (first author, publication year, study design, study setting, sample size, and participant characteristics) and outcome data into Microsoft Excel form. We solved any differences of opinion about the study characteristics by consulting a third review author (CC).

### 2.5. Quality Assessment

Two reviewers (CC, FF) independently assessed study quality, and discrepancies were resolved through discussion. We resolved any differences in opinion about the methodological quality by consulting a third review author (SN). We used the JBI checklists, where appropriate, to perform observational studies, while the Cochrane Risk of Bias tool (Rob1) was used to perform randomized controlled trials.

### 2.6. Evidence Synthesis and Statistical Analysis

All data were collected to perform the systematic reviews using Review Manager software (version 5.4.1). The number of included studies, the estimated risk of bias, and the heterogeneity were considered to decide if a meta-analysis was possible. We planned to perform two steps: (i) compare fixed and random-effects models using STATA version 17th; (ii) produce the forest plot according to a preliminary analysis of the findings. We established and used the mean difference (or standardized mean difference if different outcome measures for the same outcome were available) to set the time in the brace as a continuous outcome and examine heterogeneity using I^2^ (CI 95%) and χ^2^ tests. I^2^ > 60 was considered to represent high level of heterogeneity. We performed a narrative synthesis with frequencies if the meta-analysis was not applicable. Weighted means and standard deviations were estimated where possible.

We expected a high heterogeneity of studies, with different types of sensors being used. Therefore, we planned to subgroup studies according to the type of sensors used (i.e., tension, temperature, or movement), assuming that the number of studies was enough. We also planned to subgroup studies according to the outcome measures and the interventions made to increase compliance. We planned to compare groups using parametric or non-parametric tests where appropriate.

## 3. Results

### 3.1. Study Selection

After removing duplicates, 462 records were screened for title and abstract, of which 427 were discarded. We retrieved 34 full texts from the remaining eligible articles, assessed them, and discarded those that did not meet the inclusion criteria. Finally, 17 texts were included, and 5 publications were identified via the relevant systematic reviews. Figure 1 illustrates the flow diagram of study selection.

### 3.2. General Characteristics

According to their design, we categorized 12 studies as case series, 8 studies as observational studies, 1 as a quasi-experimental study, and 1 as a randomized control trial (RCT). Three of the retrieved studies were not useful for our purpose and were not furtherly considered when performing the subsequent analyses, since one study merely included a population of two people for validation purposes [13], one study reported data of interest in a different article already included in the present review [14], and one study only reported results in graphics, with no numerical values [15]. In addition, one study partially reported results as medians [16], while another study did not report data about compliance, only recording the brace-wearing hours [17].

A total of 1107 scoliosis patients were studied, and 85% (940) of these patients were females. The average age was 12.7 ± 1.6, according to data provided by 19 studies. The average baseline Cobb angle was 36.5 ± 10.5, which was derived from 11 studies. The Risser stage ranged between 0 and 2, being derived from six studies. The types of braces used were generic TLSO braces in 10 studies; Boston braces in 3 studies; Chêneau braces in 2 studies; Sforzesco, Sibilla, and Lapadula braces in 1 study; Sforzesco braces alone in 1 study; Spinecor braces in 1 study; and in 2 cases, both Boston and generic TLSO braces. Two studies did not state the type of brace used (Table 1). The type of sensors used were temperature sensors in 15 studies and a force transducer in 4 studies.

### 3.3. Risk of Bias Assessment and Critical Appraisal of the Included Studies

In the case series [13,15,18,20,21,25,26,27,28,31,32,34], all questions on the JBI checklist received a good percentage of positive answers, except for the clear reporting of consecutive participants’ inclusion. With regard the observational studies [14,16,17,19,23,24,30,33], the main methodological limitations were associated with the absence of the identification of confounding factors and strategies to manage them, as well as the application of strategies to address incomplete follow-up visits. The appraisal of the quasi-experimental study [22] demonstrated that the main methodological shortcomings in terms of an incomplete follow-up were not adequately analyzed. Finally, the risk of bias assessment applied to the included RCT [29] showed an unclear risk of allocation concealment and blinding of the outcome assessors, as well as a high risk of personnel blinding. Table 2 provides the results of the critical appraisal performed on the studies included in the present review.

### 3.4. Evidence Synthesis

Due to the characteristics of the studies, we did not perform a meta-analysis. We narratively reported the results, including the tabulation of the data and a summary of the evidence (Table 3 and Figure 2 and Figure 3).

Prescribed daily wearing hours were 20.6 (range 8–24). All studies bar three [22,27,31] reported this data, but the average was calculated based on nine studies [16,18,20,21,24,25,26,27,29] because the other studies only provided a range.

The weighted average number of hours recorded by the sensor in the five studies that reported such values (with standard deviations) was 12.4 ± 5.4 [17,18,21,24,33]. We calculated a weighted average of 13.1 h from 12 studies, adding 2 studies that did not report the standard deviation [16,22] and 5 studies that had information that allowed us to calculate the average wearing hours. Finally, one study reported subgroup’s information but not their population and, thus, could not be included in the weighted averages [34]. The data among subgroups ranged between 5.7 and 21 h per day (Figure 2).

Following the same methods, we found an average recorded compliance of 70.4% ± 25.9% from 12 studies [18,20,21,24,26,27,28,29,31,32,33,34], as well as 61.2% when including studies without standard deviation [16,19,23,25,30]. The range among subgroups was between 21.8 and 93.9% (Figure 3).

### 3.5. Effect of Interventions Performed to Increase Compliance

Karol tested the efficacy of counseling [22], and Miller checked the addition to counseling of the information regarding the presence of the sensor [29]: both studies found efficacy data related to these interventions. Among the other studies, Donzelli [19] and Hasler [20] reported the implementation of interventions to increase compliance in their observational studies, while all other papers either did not provide interventions or did not report the related information.

Overall, 249 patients who received interventions to increase compliance showed an adherence of 66%, compared to 58.6% in 469 patients with absent information or no intervention. The recorded brace wearing in the groups was 15.1 h (269 patients) vs. 11.9 h per day (304 participants). A statistical comparison was not possible because we did not have enough information to calculate the standard deviations in all groups.

### 3.6. Effect of Type of Sensor

Only four studies performed by the same groups reported the use of force sensors [25,26,27,28], while all other studies used temperature sensors. We did not attempt any comparison due to the low number of patients [33] in the force sensors groups.

## 4. Discussion

Based on our objectives, we found that (1) compliance greatly varied between 21.8 and 93.9% (weighted average: 58.6%), (2) real brace wearing time varied between 5.7 and 21 h per day (weighted average 13.1), and (3) interventions seemed to improve both outcomes, with compliance increasing from 58.6 to 66% and brace wearing increasing from 11.9 to 15.1 h per day. The two comparative studies showed the positive effects of a stand-alone intervention (counseling) [22] or an addictive intervention (information added to counseling) [29].

Bracing is the most effective conservative approach to AIS, but to reach its therapeutic outcomes, good patient adherence to the prescribed regimen of wearing hours is crucial [35]. Using patient reporting-based systems to monitor compliance have often been shown to lead to over-estimation; therefore, electronic monitoring systems have been developed to help improve patients’ compliance. According to a few studies, these devices have proven to be 98.5% reliable in terms of measuring wear times [18,34].

Despite the limited number of studies reporting patient-reported compliance, it is clear that there is a significant discrepancy between the recorded and reported data. In our opinion, this topic deserves further investigation through the performance of targeted studies, as it could have important implications in clinical practice. If the discrepancy is confirmed, clinicians should approach the patient-reported data with caution in the absence of sensor data and take it into account both when prescribing wearing hours and during treatment monitoring. Furthermore, this finding would support the recommendation of using sensors in clinical practice to enhance treatment monitoring of brace wearing, in addition to other key elements, like well-designed and well-built braces, as well as the commitment of the treating team [36].

As previously mentioned, many aspects need to be considered when dealing with AIS patients’ compliance. Among these aspects is prescribed wearing hours [10,36]. Based on the data provided by the studies that we analyzed, it was not possible to establish a sufficient correlation between compliance values and the number of prescribed hours to confidently confirm a difference in compliance. Takemitsu et al. [34] showed that as the number of hours prescribed by the physician increases, and the actual wearing hours of patients increases, compliance gradually decreases. This result highlights the importance of offering patient counseling at the time of prescription. Nonetheless, this topic could be worthy of further investigation.

Karol and Miller, in 2016 and 2012, respectively [22,29], investigated informative and educational approaches, in addition to sensor-based monitoring, highlighting significant differences in orthotic treatment outcomes between the population that received such interventions and the control groups. Therefore, a plausible hypothesis is that the use of sensors could contribute to improving patient compliance. However, there is a belief that the improved adherence aspect may be due to patients participating in a clinical study, rather than solely the presence of sensor technology, and it may not be reflected in routine clinical practice [37].

The relevance of proper counseling is also highlighted in the SOSORT guidelines released in 2008 [38], which summarize the different components considered to be essential for achieving a good outcome in brace usage. These components include the experience and expertise of both the medical team and the certified prosthetist/orthotist (CPO); the presence of a multidisciplinary team composed of a physician, CPO, and physiotherapist who can provide proper counseling and dedication to the patient; and, lastly, the correctness and accuracy of the orthotic prescription provided by the physician. This appropriateness of prescribing not only needs to be supported by precise technical specifications and a correct prescription regarding wearing hours, which can be tailored to the patient’s characteristics, but should also result from shared decision-making between the physician and the patient (and their family). Family members should be properly informed about possible outcomes, the risks of progression, and possible alternatives to ensure that the treatment can be a shared choice between the parties.

The prescribing physician should also outline all possible strategies that can improve patients’ compliance with orthotic treatment. One important strategy is the rewarding weaning system, which involves gradually reducing the prescribed wearing hours every six months in patients with daily regimens of 22 h or more until gradual complete weaning is achieved once the patient surpasses US Risser stage 4. Another possible strategy is the use of braces that have low visibility under clothing, which is a technical factor that, based on our clinical experience, positively influences the acceptance of the brace and the compliance of AIS patients [39].

In summary, factors that improve compliance are essential to achieving good results. The various pieces of evidence agree regarding the aforementioned medical and human factors, such as correct prescription, counseling, information, multidisciplinarity, shared decision-making, rewarding weaning systems, and technical factors. Wearable sensors can influence many of the above-mentioned items, contributing to the enhancement of the final clinical outcomes.

This study has some limitations. For example, the results are based on studies with low-quality designs: we only retrieved one RCT, and the majority of the studies did not use a control group.

Other limitations of our study include the fact that we only included studies conducted in the English and Italian languages, and we did not contact authors of articles with missing full text. We also did not include conference abstracts, as they were not peer reviewed; however, they could have contained interesting data related to our study.

The quality of the data collected may also be affected by the low sample sizes of the studies that were analyzed, with samples containing fewer than 10 participants in many cases.

Another limitation was found regarding the heterogeneity of the results reported in the studies, with many studies not reporting data that are fundamental when addressing compliance. Despite widespread reporting of measured compliance, most of the studies did not report the average number of wearing hours reported by the patient, and very few studies reported recorded wearing hours. The absence of this data makes it difficult to compare recorded compliance and patient-reported compliance to understand if sensors can be a useful tool in improving the monitoring of compliance.

It is our opinion that a better agreement regarding the data that need to be collected when addressing compliance with bracing is needed to enable further study.

Moreover, many other issues remain related to optimizing the treatment of AIS. Indeed, the lack of research in some areas and the poor quality of studies are among the challenges currently being faced in the field [40].

Poor quality research limits the possibility of increasing our knowledge and improving the treatments available to patients. Many questions still exist about the protocols, the weaning phase, the role of exercise associated with bracing and compliance, the materials used, and the methods used in brace construction.

Decades of studies regarding the effectiveness of treatments have led to the current knowledge of idiopathic scoliosis [9,38]; however, the steps required to arrive at a comprehensive approach to scoliosis are missing.

## 5. Conclusions

Sensors can be useful, simple, and affordable tools for objectively and continuously monitoring adherence to therapy in everyday clinical practice. Having an objective measure of compliance provided by the sensors allows the clinician to make informed decisions and prescribe therapy in a personalized and sustainable manner, balancing therapeutic efficacy with the patients’ daily needs and difficulties [41]. The current review suggested that greater adherence could be achieved by combining sensors’ monitoring with multimodal interventions, including educational and exercise components. However, studies of these topics are supported by low-quality evidence. Further studies are needed to understand the available data, which are more homogeneous in the reported results, which more thoroughly. investigate both the measured aspect and the aspect reported by the patient Finally, with technological knowledge rapidly advancing [42,43], it will be important to bring these tools into clinical studies to provide patients with increasingly useful, accurate, and non-invasive devices.

## Figures and Tables

**Figure 1 sensors-23-07660-f001:**
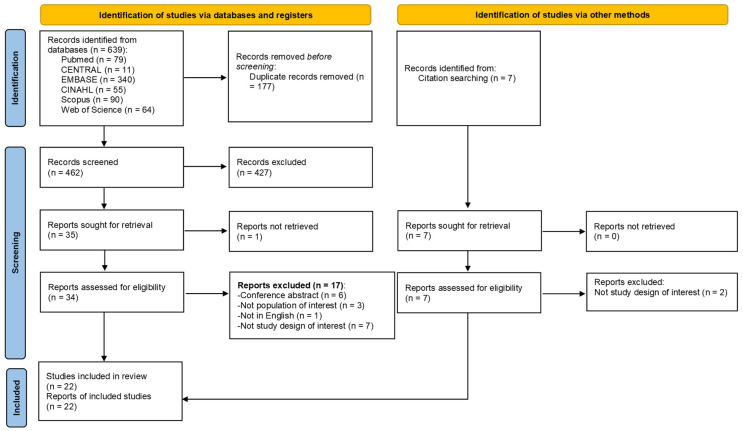
PRISMA flow diagram.

**Figure 2 sensors-23-07660-f002:**
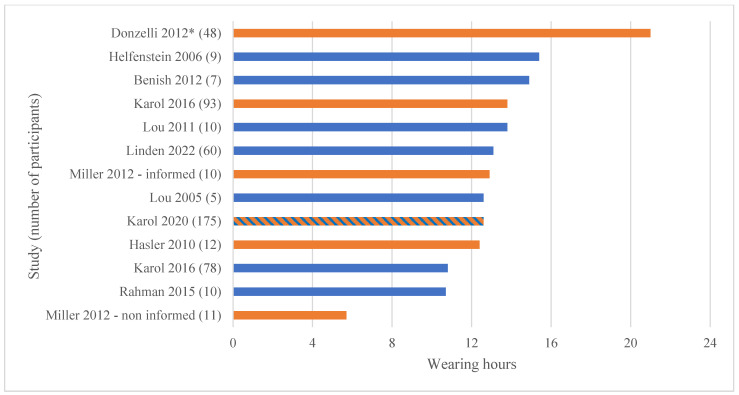
Mean recorded wearing hours reported in the included studies. Legend: orange bars represent groups that received compliance interventions (education, counseling, or multiple interventions); blue bars represent groups that did not receive compliance interventions or in which such interventions were not reported; blue–orange bar represents a mixed population in which not all participants received a compliance intervention. (*) = data reported as the median [16,17,18,20,21,22,24,26,28,29,33].

**Figure 3 sensors-23-07660-f003:**
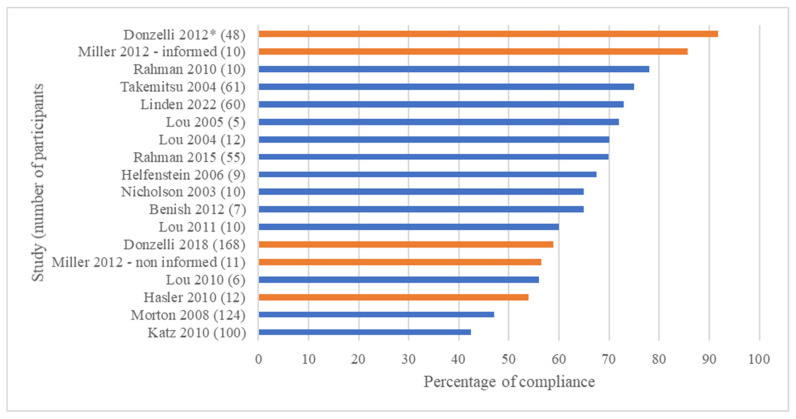
Mean recorded compliance (%) reported by the included studies. Legend: orange bars represent groups that received compliance interventions (education, counseling, or multiple interventions); blue bars represent groups that did not receive compliance interventions or in which such interventions were not reported. (*) = data reported as the median [16,18,20,21,23,24,25,26,27,28,29,30,31,32,33,34].

**Table 1 sensors-23-07660-t001:** Characteristics of the included studies.

Study	Study Design	Brace Type	Group	Sample Size	Gender	Age (Years)		Risser		Baseline Cobb (°)			
				N	M/F	Mean	SD	Mean	SD	Mean	SD	min	max
Benish 2012 [18]	Case series	TLSO		7	0/7	14.1	1						
Chalmers 2014 [15]	Case series	TLSO		7	1/6	13.7	1.7			35	6		
Donzelli 2012 [16]	Prospective cohort	Sibilla. Sforzesco. Lapadula		48	10/38	14.1	1.1	2.6	1.3	36.8	10.6		
Donzelli 2018 [19]	Retrospective cohort	Sforzesco	Adherent	104	13/91	12.9	1.8			41.7	11.2		
			Non-adherent	64	50/14	13.7	1.9			41.1	11.8		
			TOTAL	168	63/105	13.2	1.8			41.5	11.4		
Hasler 2010 [20]	Case series	Spinecor		12	0/12	13	1.4			27.4	4.6	20	35
Helfenstein 2006 [21]	Case series	Chaneau		9	0/9	14.2	1.5						
Karol 2016 [22]	Quasi-experimental	Boston and TLSO	Counseled	93	11/82			0.5	0.7	33.2			
			Non-counseled	78	6/72			0.3	0.6	33.9			
			TOTAL	171	17/154	12.3		0.4	0.7	33.5		25	45
Karol 2020 [17]	Prospective cohort	Boston and TLSO	Underweight	15		12.7		0.3	0.7	35.3			
			Normal	130		12.3		0.4	0.7	32.9			
			Overweight	21		11.9		0.2	0.5	33			
			Obese	9		12		0.3	0.7	37.1			
			TOTAL	175	16/159	12.3		0.4	0.7	33.4			
Katz 2010 [23]	Prospective cohort	Boston		100	9/91	12.7		0–1 *				25	45
Linden 2022 [24]	Retrospective cohort	Boston	Adherent	32	5/27	12.5	1.1	0–2 *					
			Non-adherent	28	3/25	12.4	1.3						
			TOTAL	60	8/52	12.5	1.2						
Lou 2004 [25]	Case series	Not reported		12	3/9	14.1	1.4			34	9		
Lou 2005 [26]	Case series	Not reported		5	1/4	12.6	2.2			31	5		
Lou 2010 [27]	Case series	TLSO		6	1/5	12.3	1			33.4	4		
Lou 2011 [28]	Case series	TSLO		10	1/9	11.6	1.3			31.8	6.6		
Miller 2012 [29]	RCT	TSLO	Informed	10	3/7	11.9	2.3	0–2 *		33	9	20	50
			Non-informed	11	2/9	12.9	1.7			29.3	7.9		
			TOTAL	21	5/16	12.4	2			31.1	8.4	20	50
Morton 2008 [30]	Prospective cohort	Boston		124	16/108								
Müller 2010 [13]	Case series	Cheneau		2	0/2								
Nicholson 2003 [31]	Case series	TLSO		10	0/10	15	1.2						
Rahman 2005 [14]	Prospective cohort	TLSO		34	4/30	12							
Rahman 2010 [32]	Case series	TLSO		10	2/8	13.3							
Rahman 2015 [33]	Prospective cohort	TLSO		55	3/52							25	40
Takemitsu 2004 [34]	Case series	TLSO		61	7/54	12						20	45
TOTAL (ponderate mean)				1107	167/940	12.7	1.6			36.5	10.5	24	44

Abbreviations: N = number of participants; M = male; F = female; SD = standard deviation; min = minimum; max = maximum; RCT = randomized controlled trial; TLSO = thoracic-lumbar-sacral orthosis. (*) = data reported as ranges. Three studies [13,14,15] were not useful for our study’s purpose and were not further considered when performing the subsequent analyses (see the text for further details).

**Table 2 sensors-23-07660-t002:** Critical appraisal of the included published studies.

Case Series	1	2	3	4	5	6	7	8	9	10	
Benish 2011 [18]	-	+	+	?	+	+	-	+	-	?	
Chalmers 2014 [15]	?	?	?	?	+	+	+	+	-	+	
Hasler 2010 [20]	+	+	+	+	-	+	+	+	+	+	
Helfenstein 2006 [21]	+	+	+	?	+	+	-	+	-	?	
Lou 2004 [25]	+	+	+	?	+	+	+	+	-	+	
Lou 2005 [26]	-	?	?	?	+	+	+	+	-	?	
Lou 2010 [27]	-	?	?	?	+	+	+	+	-	+	
Lou 2011 [28]	+	+	+	?	+	+	+	+	-	+	
Müller 2010 [13]	-	?	?	?	+	+	+	+	-	?	
Nichols 2003 [31]	+	+	+	?	+	+	-	+	-	?	
Rahman 2010 [32]	+	+	+	?	+	+	+	+	-	?	
Takemitsu 2004 [34]	+	+	+	+	-	+	+	+	+	+	
Cohort studies	1	2	3	4	5	6	7	8	9	10	11
Donzelli 2012 [16]	+	+	+	-	-	+	+	+	+	?	+
Donzelli 2018 [19]	-	+	+	+	+	+	+	+	+	+	+
Karol 2020 [17]	+	+	-	-	+	+	+	+	+	?	+
Katz 2010 [23]	?	+	+	-	-	+	+	+	-	-	+
Linden 2022 [24]	-	-	+	-	-	+	+	+	+	?	+
Morton 2008 [30]	+	+	+	-	-	+	+	+	+	?	+
Rahman 2005 [14]	?	?	+	-	-	+	-	+	+	?	?
Rahman 2015 [33]	-	-	+	-	-	+	+	+	-	-	?
Quasi-experimental	1	2	3	4	5	6	7	8	9		
Karol 2016 [22]	+	+	+	+	+	-	+	+	+		
RCT	1	2	3	4	5	6	7	8	9		
Miller 2012 [29]	+	?	+	-	?	-	+	+	-		

Case series items: (1) Were there clear criteria for inclusion in the case series? (2) Was the condition measured in a standard and reliable way for all participants included in the case series? (3) Were valid methods used for the identification of the condition for all participants included in the case series? (4) Did the case series have a consecutive inclusion of the participants? (5) Did the case series have complete the inclusion of the participants? (6) Was there clear reporting of the demographics of the participants in the study? (7) Was there clear reporting of the clinical information of the participants? (8) Were the outcomes or follow-up results of cases clearly reported? (9) Was there clear reporting of the presenting site(s)/clinic(s) demographic information? (10) Was statistical analysis appropriate? Cohort studies items: (1) Were the two groups similar and recruited from the same population? (2) Were the exposures measured similarly to assign people to both exposed and unexposed groups? (3) Was the exposure measured in a valid and reliable way? (4) Were the confounding factors identified? (5) Were strategies used to deal with the confounding factors stated? (6) Were the groups/participants free of the outcome at the start of the study (or at the moment of exposure)? (7) Were the outcomes measured in a valid and reliable way? (8) Was the follow-up time reported to be sufficient to be long enough for outcomes to occur? (9) Was a follow-up completed, and if not, were the reasons for a lack of a follow-up described and explored? (10) Were the strategies used to address the incomplete follow-up utilized? (11) Was appropriate statistical analysis used? Quasi-experimental studies items: (1) Is it clear in the study what represents the “cause” and what represents the “effect”? (2) Were the participants who were included in any comparisons similar? (3) Did the participants who were included in any comparisons receive similar treatment/care, other than the exposure or intervention of interest? (4) Was there a control group? (5) Were there multiple measurements of the outcome both pre- and post-intervention/exposure? (6) Was the follow-up completed and, if not, were the differences between the groups in terms of their follow-up adequately described and analyzed? (7) Were the outcomes of the participants included in any comparisons measured in the same way? (8) Were the outcomes measured in a reliable way? (9) Was the appropriate statistical analysis used? Risk of bias assessment items: (1) generation of allocation sequence; (2) allocation concealment; (3) blinding of participants; (4) blinding of personnel; (5) blinding of outcome assessors; (6) incomplete outcome data; (7) selective reporting; (8) other bias—group similarity; (9) other bias—intention to treat analysis. Abbreviations: “+”—yes/low risk of bias; “-“—no/high risk of bias; “?”—unclear.

**Table 3 sensors-23-07660-t003:** Data provided by each study about sensors, wearing time, and compliance.

	Sensor Type	Interventions to Increase Compliance	Sample Size	Prescribed Wearing Hours	Recorded Wearing Hours		Reported Wearing Hours		Recorded Compliance		Reported Compliance	
			N	Range	Mean	SD	Mean	SD	Mean%	SD	Mean%	SD
Benish 2012 [18]	Temperature	No/NR	7	23	14.9	5.9	15.1	5.9	65	25.7	65	
Donzelli 2012 [16]	Temperature	Multiple	48	23	21 **		23 **		91.7 **		100	
Donzelli 2018 [19]	Temperature	Multiple	168	18–23					58.9			
Hasler 2010 [20]	Temperature	Education	12	23	12.4	5.1			54	22.3		
Helfenstein 2006 [21]	Temperature	No/NR	9	23	15.4	4			67.5	29	94.5	
Karol 2016 [22]	Temperature	Counseled	93		13.8							
		No/NR	78		10.8							
		TOTAL	171		12.4							
Karol 2020 [17]	Temperature	Partially counseled	175	16–23	12.6	5.9						
Katz 2010 [23]	Temperature	No/NR	100	16–23					42.4			
Linden 2022 [24]	Temperature	No/NR	60	18	13.1	3.6			73	20		
Lou 2004 [25]	Force	No/NR	12	22.3 ± 1.3 *					70			
Lou 2005 [26]	Force	No/NR	5	17.5 ± 3.8 *	12.6				72	15		
Lou 2010 [27]	Force	No/NR	6						56	15		
Lou 2011 [28]	Force	No/NR	10	23	13.8	2.1			60	11.9		
Miller 2012 [29]	Temperature	Multiple—informed	10	18	15.4	4.8	15		85.7	26.5		
		Counseled—non-informed	11		10.2	5.4	10		56.5	30.2		
		TOTAL	21		12.7	5.2	12.4		70.4	29		
Morton 2008 [30]	Temperature	No/NR	124	16–23					47			
Nicholson 2003 [31]	Temperature	No/NR	10						65	25	89	
Rahman 2010 [32]	Temperature	No/NR	10	8 or 12					78	28		
Rahman 2015 [33]	Temperature	No/NR	55	8–24	10.7	5.2			69.9	31.5		
Takemitsu 2004 [34]	Temperature	No/NR	61						75	27	85	24
TOTAL (ponderate mean)			871		13.5				61.2			
With compliance interventions		N wearing hours/compliance	269/249		15.6				66.0			
Without or not reported compliance interventions		N wearing hours/compliance	304/469		11.9				58.6			

Abbreviations: SD = standard deviations; % = percentage; N = number of participants; NR = not reported; Compliance was considered to be the ratio of reported/recorded wearing hours to prescribed wearing hours. (*) = data reported as the mean and standard deviation; (**) = data reported as the median. Three studies [13,14,15] were not useful to our study and were not considered during the subsequent analyses (see text for further details). One study [17] was not included in the analyses of interventions performed to increase compliance since it included a mixed population.

## Data Availability

Data is contained within the article.

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
