# Peer review of "Influence of Specific Interventions on Bracing Compliance in Adolescents with Idiopathic Scoliosis—A Systematic Review of Papers Including Sensors’ Monitoring"

_sensors, 2023, doi:10.3390/s23177660_

Round 1

Reviewer 1 Report

Current manuscript entitled “Specific interventions can increase compliance to bracing in adolescents. results from a systematic review of papers including sensors’ monitoring” deliberated on the compliance of adolescents with AIS to their brace treatment, find out how long they wear it through objective electronic monitoring and verify if specific interventions can increase adherence. Wearable sensors are available to objectively monitor the brace-wearing time. The manuscript seems good and can be accepted after addressing the following comments.

1.      Authors should consider revising the title of the manuscript.

2.      Clear statements of the novelty of the work should also appear briefly in the Abstract and Conclusions sections.

3.      The Abstract should contain answers to the following questions: What problem was studied and why is it important? What methods were used? What are the important results? What conclusions can be drawn from the results? What is the novelty of the work and where does it go beyond previous efforts in the literature? Add the main findings and objective of the current study in the abstract.

4.      In the introduction, please discuss and compare the existing review articles.

5.      Provide the challenges that are currently facing with the Adolescent idiopathic scoliosis.

 Minor editing of English language required

Author Response

Current manuscript entitled “Specific interventions can increase compliance to bracing in adolescents. Results from a systematic review of papers including sensors’ monitoring” deliberated on the compliance of adolescents with AIS to their brace treatment, find out how long they wear it through objective electronic monitoring and verify if specific interventions can increase adherence. Wearable sensors are available to objectively monitor the brace-wearing time. The manuscript seems good and can be accepted after addressing the following comments.

We thank the Reviewer for the positive feedback. We described below the changes made to address his valuable comments.

Authors should consider revising the title of the manuscript.

Following the Reviewer’s suggestion, we changed the title to “Influence of specific interventions on bracing compliance in adolescents with idiopathic scoliosis. A systematic review of papers including sensors’ monitoring”. We hope that the new title can result more adequate for the manuscript.

Clear statements of the novelty of the work should also appear briefly in the Abstract and Conclusions sections.

We thank the Reviewer for the comment. As requested, we specified the current knowledge gap at the beginning of the abstract (page 1, lines 17-18) and clarified this aspect in the conclusions section (page 16, lines 456-458).

The Abstract should contain answers to the following questions: What problem was studied and why is it important? What methods were used? What are the important results? What conclusions can be drawn from the results? What is the novelty of the work and where does it go beyond previous efforts in the literature? Add the main findings and objective of the current study in the abstract.

As requested by the Reviewer we modified the objectives reported in the abstract in order to make them clearer for the readers. We also added some more details about the main findings (page 1, lines 19-25 and 33-34). Moreover, as reported in the previous comment, we underlined the current knowledge gap in the literature (page 1, lines 17-18).

Information about the studied population and methodology was already reported in the original abstract.

We thank the Reviewer for the useful comment that helped us to improve the readability of the abstract.

In the introduction, please discuss and compare the existing review articles.

During the review process, we identified two relevant systematic reviews dealing with adherence in idiopathic scoliosis (Li, X. et al. Which Interventions May Improve Bracing Compliance in Adolescent Idiopathic Scoliosis? A Systematic Review and Meta-Analysis. PLoS One 2022; Rahimi, S. et al. Effective Factors on Brace Compliance in Idiopathic Scoliosis: A Literature Review. Disabil Rehabil Assist Technol 2020) and presented some of their findings in the introduction. Following the Reviewer's comment, we modified the section to highlight the comparison of these two studies (page 2, lines 66-75).

Provide the challenges that are currently facing with the Adolescent idiopathic scoliosis.

An accurate assessment of adherence to therapy and methods to improve are crucial to optimize the treatment of adolescent idiopathic scoliosis.  Considering the great heterogeneity of the present data and the presence of few high-quality trials, this should be a focus for future studies.

Indeed, the lack of research in some areas and the poor quality of studies are among the challenges currently being faced in the field of adolescent idiopathic scoliosis. (Idiopathic Scoliosis: Novel Challenges for Researchers and Clinicians, Children- editorial)

Poor quality research limits the possibility of increasing our knowledge and improving the treatments available for our patients.

Many questions remain open about the protocols, the weaning phase, the role of exercise associated with brace and the compliance, the materials used, and the methods of brace construction.

Decades of studies on the effectiveness of treatments have led to the current knowledge of idiopathic scoliosis (Establishing Consensus on the Best Practice Guidelines for the Use of Bracing in Adolescent Idiopathic Scoliosis. Spine Deform. 2020; Guidelines on “Standards of Management of Idiopathic Scoliosis with Corrective Braces in Everyday Clinics and in Clinical Research”: SOSORT Consensus 2008. Scoliosis 2009), however the steps to arrive at a comprehensive approach to scoliosis are missing.

Following the Reviewer's suggestion, we added these considerations to the final part of the discussion section (pages 15-16, lines 438-447)

Minor editing of English language required

We thank the Reviewer for the helpful suggestion. We ran a language check using professional software for English writing (Grammarly, Grammarly Inc., San Francisco, CA, USA).

Reviewer 2 Report

Sensor-based monitoring of brace wearing time has been shown to be promising tools for objectively and continuously monitoring in everyday application. This paper reviewed the impact of specific interventions on compliance with brace therapy in adolescents with AIS. The literature reviews many related articles and analyzes the useful data and information in it. I believe that this review is comprehensive and reliable, it can attract wide attention and have a certain guiding significance in this field. Therefore, it can be accepted after addressing the following problems:

 1、  The authors should consider whether an additional paragraph is necessary to describe the workings or design of sensors in electronic monitoring of brace compliance.

2、  The conclusion about the role of sensor technology in influencing compliance outcomes is unclear, which is important for the integrity of this article. I suggest that some additional content be added in item 5-Conclusions or 4-Disscussion to address this issue.

3、  In item 5-Conclusions, I suggest that the authors combine these two paragraphs.

4、  Some recent works on health monitoring sensors, for example, Smart Health, 2021, 21, 100179; Nano Research, 2023, 16(1): 1196-1204, should enrich the background and help improve the manuscript.

5、  The author should add more perspective discussion in the Conclusion part.

Author Response

Sensor-based monitoring of brace wearing time has been shown to be promising tools for objectively and continuously monitoring in everyday application. This paper reviewed the impact of specific interventions on compliance with brace therapy in adolescents with AIS. The literature reviews many related articles and analyzes the useful data and information in it. I believe that this review is comprehensive and reliable, it can attract wide attention and have a certain guiding significance in this field. Therefore, it can be accepted after addressing the following problems:

We thank the Reviewer for the positive comments. In the following lines, we tried to address all the issues raised to improve the manuscript.

The authors should consider whether an additional paragraph is necessary to describe the workings or design of sensors in electronic monitoring of brace compliance.

We thank the Reviewer for the suggestion. Considering the clinical nature of the manuscript, we decided not to introduce a specific paragraph on the technical properties of the sensors. However, we provided throughout the text some specific references on the type of sensors applied in clinical contexts as well as their reliability (Rahimi, S. et al. Effective Factors on Brace Compliance in Idiopathic Scoliosis: A Literature Review. Disabil Rehabil Assist Technol 2020; Benish, B.M. et al. Validation of a Miniature Thermochron for Monitoring Thoracolumbosacral Orthosis Wear Time. Spine 2012; Takemitsu, M. et al. Compliance Monitoring of Brace Treatment for Patients with Idiopathic Scoliosis. Spine 2004).

The conclusion about the role of sensor technology in influencing compliance outcomes is unclear, which is important for the integrity of this article. I suggest that some additional content be added in item 5-Conclusions or 4-Disscussion to address this issue.

As suggested by the Reviewer, we modified the conclusions section underlining the importance of objective measurement through sensors to make clinically informed decisions and prescribe therapy in a personalized and economically sustainable manner for the patient, balancing therapeutic efficacy with their daily needs and difficulties (page 16, lines 449-456). In addition, as suggested by other comments, we specified that sensor-monitored bracing, combined with other specific interventions, could result in even greater benefits (page 16, lines 456-458). Finally, we also introduce an innovative perspective on the use of very recent biomaterials in this clinical field (page 16, lines 463-467).

 In item 5-Conclusions, I suggest that the authors combine these two paragraphs.

We thank the Reviewer for the useful comment. We combined the conclusion section in a single paragraph.

Some recent works on health monitoring sensors, for example, Smart Health, 2021, 21, 100179; Nano Research, 2023, 16(1): 1196-1204, should enrich the background and help improve the manuscript.

We thank the Reviewer for the interesting suggestion. Considering the novelty of the topics tackled by the articles, we included them in the final part of the discussion in order to stimulate further research and debate on these issues (page 16, lines 463-467).

The author should add more perspective discussion in the Conclusion part.

We thank the Reviewer for the comment. As suggested and reported in the previous point, we modified the conclusions section and added a final statement on future studies considering the most recent advancement in the field of wearables technologies (page 16, lines 460-467).

Reviewer 3 Report

i am grateful for the opportunity to review your manuscript. 

i offer the following for your consideration:

- the title of your manuscript might be re-phrased to "the influence of specific interventions such as the use of sensors on bracing in adolescents: a systematic review" (or similar);

- in the final paragraph of section 2, please express "not parametric" to "non-parametric";

- for Figure 2, the caption is presently mis-wrapped against the lower right corner of the figure itself.

i would like to invite you to do your best to find a native English speaker to proofread your work, not so much to catch errors of spelling or of grammar, but instead to look out for idiomatic expressions and tonal inconsistencies, such as the present phrasing of the title of the manuscript.

Author Response

I am grateful for the opportunity to review your manuscript. I offer the following for your consideration:

We thank the Reviewer for the useful comments delivered.

The title of your manuscript might be re-phrased to "the influence of specific interventions such as the use of sensors on bracing in adolescents: a systematic review" (or similar);

As requested also by Reviewer 1, we change the title to “Influence of specific interventions on bracing compliance in adolescents with idiopathic scoliosis. A systematic review of papers including sensors’ monitoring”. We focused the attention on “papers including sensors’ monitoring” in the title since specific interventions (i.e. educational, and counseling sessions as well as exercises) have been already described in the literature without objective sensors measurements

In the final paragraph of section 2, please express "not parametric" to "non-parametric";

As requested, we changed “not parametric” to “non-parametric” at page 4, line 169.

For Figure 2, the caption is presently mis-wrapped against the lower right corner of the figure itself.

We thank the Reviewer for the comment. We corrected the graphical issue and moved the caption under Figure 2.

I would like to invite you to do your best to find a native English speaker to proofread your work, not so much to catch errors of spelling or of grammar, but instead to look out for idiomatic expressions and tonal inconsistencies, such as the present phrasing of the title of the manuscript.

We thank the Reviewer for the useful suggestion. As also requested by Reviewer 1, we ran a language check with professional software for English writing (Grammarly, Grammarly Inc., San Francisco, CA, USA). We also changed the title of the manuscript.